# The Chloroplast Genome of the Lichen Photobiont *Trebouxiophyceae* sp. DW1 and Its Phylogenetic Implications

**DOI:** 10.3390/genes13101840

**Published:** 2022-10-12

**Authors:** Lidan Wang, Shenglu Zhang, Jinjin Fang, Xinjie Jin, Reyim Mamut, Pan Li

**Affiliations:** 1College of Life Sciences and Technology, Xinjiang University, Urumchi 830046, China; 2College of Life Sciences, Zhejiang University, Hangzhou 310058, China; 3College of Life and Environmental Science, Wenzhou University, Wenzhou 325035, China

**Keywords:** algae, genome size, *Peltigera*, Prasiolales, phylogeny

## Abstract

Lichens are symbiotic associations of algae and fungi. The genetic mechanism of the symbiosis of lichens and the influence of symbiosis on the size and composition of the genomes of symbiotic algae have always been intriguing scientific questions explored by lichenologists. However, there were limited data on lichen genomes. Therefore, we isolated and purified a lichen symbiotic alga to obtain a single strain (*Trebouxiophyceae* sp. DW1), and then obtained its chloroplast genome information by next-generation sequencing (NGS). The chloroplast genome is 129,447 bp in length, and the GC content is 35.2%. Repetitive sequences with the length of 30–35 bp account for 1.27% of the total chloroplast genome. The simple sequence repeats are all mononucleotide repeats. Codon usage analysis showed that the genome tended to use codon ending in A/U. By comparing the length of different regions of Trebouxiophyceae genomes, we found that the changes in the length of exons, introns, and intergenic sequences affect the size of genomes. Trebouxiophyceae had an unstable chloroplast genome structure, with IRs repeatedly losing during evolution. Phylogenetic analysis showed that Trebouxiophyceae is paraphyletic, and *Trebouxiophyceae* sp. DW1 is sister to the clade of *Koliella longiseta* and *Pabia signiensis*.

## 1. Introduction

Green plants, which include green algae and embryonic plants, are one of the most important primary producers on Earth [1]. Green algae are ubiquitous in the world’s marine and freshwater ecosystems and play an important role in adapting to changing environmental conditions [2]. Green algae can become symbiotic with other species such as animals, bacteria, fungi, and plants to achieve mutualism [3,4,5]. It is estimated that green algae originated more than 1.8 billion years ago and split early into two lineages: the Charophyta and the Chlorophyta [6,7,8]. In Chlorophyta, most taxonomic advances have been made at the level of species and genera, but there are few studies above this taxonomic level and there is often a lack of consensus [9,10,11]. In the early days, Chlorophyta was traditionally divided into four groups, Ulvophyceae, Trebouxiophyceae, Chlorophyceae and Prasinophyceae, according to the characteristics of flagellar apparatus configuration and cell division process [12,13]. Later on, based on molecular data, morphology and ecological diversity, the core Chlorophyta was divided into three major classes, Ulvophyceae, Trebouxiophyceae and Chlorophyceae (UTC), plus two smaller lineages, Chlorodendrophyceae and Pedinophyceae [14,15,16]. Among the three major classes, Chlorophyceae has five orders (Oedogoniales, Chaetophorales, Chaetopeltidales, Volvocales, and Sphaeropleales), and their phylogeny are well-understood. In contrast, the phylogenic relationships within Ulvophyceae and especially Trebouxiophyceae remain unresolved [17]. Trebouxiophyceae are widely distributed throughout the world. It was thought to have seven major lineages [18]; however, as more and more new species are discovered, it is now shown that Trebouxiophyceae has as many as 16 distinct lineages [1,19,20]. Phylogenetic establishment of Trebouxiophyceae by 18s RNA and *rbc*L, however, is difficult to solve with limited data and a few gene loci [20,21]. Of the 40 or so green algae found in Trebouxiophyceae, 22 are known to form lichen clusters [22]. With the wide application of genome in phylogeny, organelle genomes have been widely used for phylogenetic construction, and it has become a common practice for the phylogenetic analysis of the entire genome (phylogenomics) [23]. From algae to higher plants, chloroplast genomes have been used to study phylogeny, providing researchers with new ideas and insights. Genome-level analysis has led to profound changes in our understanding of the evolution of green algae and has greatly improved our understanding of the deepest relationships in the phylogeny of green algae [24,25,26,27].

The chloroplast genome data from Trebouxiophyceae are very limited compared with higher plants, and its chloroplast genome has a highly variable structure [28]. Usually, the chloroplast genome is made up of four parts: the large single copy region (LSC), the small single copy region (SSC), and the two reverse repeats (IR_A_ and IR_B_). However, some algal genomes show different characteristics, including the deletion/shrinkage–disapperance/expansion of IR regions, gene rearrangement, and large intergenic spacers [29]. In previous studies, only one chloroplast genome was reported from a *Trebouxia* lichen phycobiont [30], which made it difficult to further study the structure of symbiotic algae.

In this study, symbiotic alga was isolated from *Peltigera rufescens* and then sequenced by high-throughput sequencing for the first time. The complete chloroplast genome sequence of *Trebouxiophyceae* sp. DW1 was reported, and its size, structure and gene content were compared with other published genomes of green algae. Phylogenetic analysis of Trebouxiophyceae was conducted based on 36 chloroplast genes, including 20 associated with photosynthesis, 14 ribosomal genes, and also *tuf*A and *ycf*3 genes.

## 2. Materials and Methods

### 2.1. Phycobiont Isolation and Culture Conditions

*Trebouxiophyceae* sp. DW1 was isolated from the lichen *P. rufescens* (Figure 1), the sample of *P. rufescens* was collected from Bayi Forest Farm, in Xinjiang Province. This voucher specimen was deposited in the Herbarium of College of Life Science and Technology at Xinjiang University in Urumchi, China, under the voucher number BY201830, by grinding the sterilized lichens and culturing it in BG11 medium [31] in a growth chamber at 22 °C under a 10:14 h light:dark cycle.

### 2.2. DNA Extraction, Sequencing and Raw Data Preprocessing

DNA was extracted using the CTAB method [32]. The sequencing was performed in Hiseq PE150 mode (pair-end sequencing). Before the quality control, the original data were evaluated with BBTools [33] statistical information, and some basic information was counted and visualized to determine the data quality. In order to improve the quality and reliability of the follow-up data analyses, we carried out quality control on the original data. Using BBduk [33], to remove possible joint sequences in reading sequence; quality trimming: sliding window method to remove continuous low-quality sequences (Q value < 20) at both ends of reading sequence; to remove reading sequence and its paired reading sequence with length less than 35 bp after quality control. To evaluate the pollution of reading sequences we compared BLAST+, using 10,000 sequences randomly selected from QC (quality control). The comparison database was NCBI NT. The results of comparison were evaluated with e value ≤ 1 × 10^−10^ and similarity > 90%, coverage > 80%. This step of analysis is to determine whether the sample is contaminated during the preparation and sequencing of the library. Possible contamination may include microbial contamination of the environment, and contamination of human genetic material that may be introduced into the laboratory operation, etc.

### 2.3. Genome Assembly

SPAdes (http://bioinf.spbau.ru/SPAdes/) (accessed on 11 September 2020) was first used to correct the sequence errors of original sequence and then to assemble by multiple Kmer values. Finally, the best results were obtained by synthesizing the Kmer values (K-mer = 39). Therefore, the assembly effect and accuracy of SPAdes were high. Gapcloser v1.12 [34] and GapFiller v2.1.2 [35], the gap-filling tools, use a large number of pairs of short sequences to complete the gap-filling process.

### 2.4. Gene Prediction

The chloroplast genomes were annotated by GeSeq (https://chlorobox.mpimp-golm.mpg.de/geseq.html) (accessed on 27 January 2021) [36] and then the wrong genes were manually corrected by Sequin software, which manually adjusts the position of start and stop codons. Lastly, the results were placed in OGDRAW (https://chlorobox.mpimp-golm.mpg.de/OGDraw.html) (accessed on 23 March 2021) [36] to generate the cpDNA map.

### 2.5. Codon Usage

With the exception of Met and Trp, amino-acid residues are encoded by two or more synonyms, and genes have a tendency to use synonyms nonrandomly to encode amino acids, which is known as codon usage bias. Codon usage bias analysis could provide clues for revealing the law of genetic evolution [37]. Therefore, we analyzed codon usage bias of *Trebouxiophyceae* sp. DW1 in Mega v7.0.26 [38].

### 2.6. Identification of Repeat and Simple Sequence Repeat

Repeat sequences were detected with the online version of REPuter [39]. We included forward and palindromic repeat sequences with a minimum repeat size of 30 bp. Meanwhile, online Tandem Repeats Finder was used to search for Tandem Repeats in chloroplast DNA sequences [40]. Simple sequence repeats (SSR) were performed by using MISA (https://webblast.ipk-gatersleben.de/misa/) (accessed on 19 December 2020) for prediction, and the parameter was set as follows: the minimum SSR sequence segment length of 10 bp [41].

### 2.7. Phylogenetic Analysis

Thirty-six chloroplast genes (atpA, atpB, atpE, petA, petB, psaA, psaB, psbA, psbB, psbC, psbD, psbE, psbH, psbJ, psbK, psbL, psbN, psbT, psbZ, rbcL, rpl2, rpl5, rpl14, rpl16, rpl20, rpl23, rps2, rps3, rps7, rps8, rps11, rps12, rps18, rps19, tufA and ycf3) from 22 species of green algae were downloaded from GenBank. These sequences, together with those of *Trebouxiophyceae* sp. DW1, were used for phylogenetic analysis (Table 1). The sequences were aligned with MAFFT version 7 [42] and trimmed with Gblocks version 0.91b [43]. Maximum-likelihood (ML) method was used to construct phylogenetic tree by RAxML v7.2.8 [44], bootstrap repeats were 1×10^6^ generations, and the best model was selected with ModelFinder [45]. FigTree v3.2 was used to exhibit the consensus tree.

## 3. Results

### 3.1. Structural Characteristics of Chloroplast Genome

The length of *Trebouxiophyceae* sp. DW1 chloroplast genome is 129,447 bp, of which the LSC is 98,101 bp, the SSC is 16,602 bp, and the two inverted repeats (IR_A_ and IR_B_) are 7372 bp (Figure 2). The chloroplast genome of *Trebouxiophyceae* sp. DW1 contains 103 genes (Appendix A), including 67 protein-coding genes, 32 transport RNAs (tRNAs), and 2 ribosomal RNAs (rRNAs) with exons and intergenic sequences occupy a large proportion (Figure 3). *Rpo*C2 has two introns, while *rpo*B, *rpo*C1, and *trn*L(uaa) each have one intron. The length of the introns ranged from 404 to 1574 bp. The IRs region contains one protein-coding gene *pet*D, two tRNAs and two rRNAs, among which *rrn*23 is a pseudogene.

### 3.2. Repeat Elements in the Trebouxiophyceae sp. DW1 Chloroplast Genome

A total of 49 repetitive sequences were detected in the chloroplast genome of *Trebouxiophyceae* sp. DW1, including 30 palindromic and 19 forward (Table 2). The size of these repetitive sequences ranged from 30 bp to 50 bp. The longest repetitive sequences were located between the *rps*7 and *ycf*4 genes. A total of 1645 bp repetitive sequences were detected in the *Trebouxiophyceae* sp. DW1 chloroplast genome, which accounted for 1.27% of the whole chloroplast genome.

Only one kind of SSRs, the mononucleotide repeats, was detected for the *Trebouxiophyceae* sp. DW1 chloroplast genome. We counted all SSRs with lengths greater than 10 bp. In the chloroplast genome of *Trebouxiophyceae* sp. DW1 (Appendix A), the main single nucleotide (p1) was A/T (about 94.74%), and no C/G was detected. Similar to other plant chloroplast genomes reported previously, simple repetitive sequences were mainly by poly thymine nucleotides (polyT) or poly adenine nucleotides (polyA), which rarely appear in a series of cytosine and guanine repeats.

We also performed tandem repeats analysis in the chloroplast genome of *Trebouxiophyceae* sp. DW1 (Table 3), which accounted for 4.64% of the whole chloroplast genome. The longest tandem sequence in the *Trebouxiophyceae* sp. DW1 chloroplast genome is 77 bp, which is located in the intergenic region between neighboring genes *pet*G and *pet*A. The highest copy numbers detected were 2.1 and 2.8 in the *Trebouxiophyceae* sp. DW1 chloroplast genome.

### 3.3. Codon Usage Analysis

Codon usage analysis indicated that TTT (for phenylalanine; Phe), AAA (for lysine; Lys), ATT (for isoleucine; Ile), CAA (for glutamine; Gln), and TTA/TTG (for leucine; Leu) in the chloroplast genome of *Trebouxiophyceae* sp. DW1 were used more frequently (Figure 4). The genomic codons have 32 codons with relative synonymous codon usage (U_RSC_) > 1, of which 28 codons ended in A/U, indicating that the chloroplast genome prefers to use codons ending in A/U.

We calculated the GC content at positions 1, 2, and 3 of the codons and recorded them as GC_1_, GC_2_, and GC_3_, respectively. GC_1_ is 35.25%, GC_2_ is 35.20%, and GC_3_ is 34.26%. The value of GC_3_ is slightly lower than GC_1_ and GC_2_. The high frequency of A and T in the codons contributes to the high AT content of the *Trebouxiophyceae* sp. DW1 chloroplast genome (64.80%).

### 3.4. Phylogenetic Analysis

Phylogeny of Trebouxiophyceae and relatives was reconstructed based on an alignment of 23,288 nt (the DNA sequences of 36 protein-coding genes), with two Streptophyta (*C. atmophyticus* and *C. vulgaris*) as outgroups (Figure 5). Trebouxiophyceae is paraphyletic, with Chlorellales more closely related with Pedinophyceae (BS = 93). *Trebouxiophyceae* sp. DW1 is sister to the clade of *K. longiseta* and *P. signiensis*, with high support (BS = 90).

## 4. Discussion

### 4.1. Features That Affect the Size of the Chloroplast Genome

To date, 153 chloroplast genomes of Chlorophyta have been published in NCBI. Among them, *Haematococcus lacustris* UTEX 2505 has the largest genome, with a length of 1,352,306 bp [46], while *Prototheca zopfii* strain SAG 2021 has the smallest genome, with a length of 28,638 bp [47]. The size of chloroplast genomes in Chlorophyta varies greatly. We compared proportions of exons, introns, and intergenic sequences and the proportions of IRs, LSC and SSC regions in some algae chloroplast genomes (Figure 6). *Trebouxia* sp. TR9 and *Prasiolosis* sp. SAG 84.81 have a larger size than others; the proportions of their exons, intergenic spacers and introns in the genomes were similar. The difference is that the latter lacks the complete chloroplast four-part chloroplast structure. *Trebouxiophyceae* sp. DW1 has a smaller genome. By comparing the genomic data, we found that the reasons for the differences in genome length variation are mainly as follows: (1) differences in the number and size of genes; (2) the length of the introns; (3) the size of the intergenic spacers; and (4) the expansion/contraction or absence of IR regions. This conclusion is consistent with previous studies [48,49,50].

There are two hypotheses about genome size. One hypothesis is the adaptive theory and the other is the junk DNA theory. The adaptive theory says that a very small genome may favor organisms with a short reproductive cycle, but a large genome may be more acceptable if the reproductive cycle is long, and changes in genome size reflect the adaptive needs of different organisms or the effectiveness of natural selection. The junk DNA theory states that extra DNA is indeed superfluous and useless so that maladaptive DNA is fixed by random drift and is passively carried on the chromosomes [51,52,53]. As the genome size increases, the ratio of insertion-sequence to coding-sequence size increases more rapidly [54,55]. The final genome size is set at a tolerable maximum, depending on the specific ecological and developmental needs of the organism. At present, the research mainly addresses the process of the evolution of genome size, and more experimental data are needed to explore the functional significance of genome size. Both genome expansion and contraction have been proposed as an evolutionary strategy to achieve the optimal balance between genomic stability and plasticity [53]. In *Trebouxiophyceae* sp. DW1, compared with the longer green-algae genome, there are fewer repeats with shorter gene segments, which may be the reason why *Trebouxiophyceae* sp. DW1 has a smaller genome.

### 4.2. Variations in Chloroplast Genome Structure

Structurally, variations are usually due to the expansion, contraction, or absence of the IRs. IRs are hot spots of genomic instability in prokaryotes and eukaryotes [56]. In eukaryotes, they break dsDNA and make chromosomes brittle, stimulate homologous recombination, and induce total chromosomal rearrangements [57,58,59,60,61]. IRs plays an important role in genetic instability, gene splicing, and replication delay, and there is a tendency to reduce genomic stability in various organisms [62,63,64,65,66].

*Trebouxiophyceae* sp. DW1 has a complete four-part structure including one SSC, one LSC, and two IRs, while *Coccomyxa* sp., *B. braunii*, *M. israelensis* and other algae lost IRs. The high propensity of IR in size, gene content and gene order, and the repetitive loss is experienced during the evolution of the Trebouxiophyceae [67,68]. Most of the IR-less genomes are rearranged, and the sequence evolves faster. The chloroplast genomes of Trebouxiophyceae show great plasticity [28,69]. Higher plants usually have complete four parts; however, IRs loss has also occurred in a few gymnosperms and angiosperms, such as *Cryptomeria japonica*, *Cephalotaxus wilsoniana*, *Taiwania cryptomerioides, Carnegiea gigantea* and *Wisteria* [70,71,72,73]. Longer IRs generally increase the stability of the genome, whereas the shrinkage and loss of IRs generally increase the likelihood of gene rearrangement and gene loss, affecting genome size [74,75]. The IRs of algae were smaller than higher plants, which tend to gradually degenerate. Perhaps IRs were not a necessary part of the chloroplast energy of algae.

### 4.3. Repeat Elements: Significant Mononucleotide Repeats in SSRs

SSRs are widely found in prokaryotes and eukaryotes. They are inevitable and highly variable products of genome replication. They exist in coding and non-coding regions of the genome and play an important role in genome evolution and recombination [76]. They were thought to participate in the gene expression, regulation, and function of components as transcription activation [77].

The 19 SSRs detected in *Trebouxiophyceae* sp. DW1 mainly existed in the non-coding region, and there was only one complex polynucleotide, and the rest were all mononucleotide. The most significant repetition pattern in the Chlorophyta chloroplast genome was mononucleotide poly A/T repetition [78], which was consistent with our results. The composition of SSRs varies greatly among different species. It was thought in the past that plants were mainly composed of trinucleotide repeats; differently, in angiosperms mononucleotide repeats were more prominent, followed by dinucleotide repeats, such as *Panax ginseng*, *Elodea canadensis*, *Helianthus annuus*, and *Olea europaea* [79,80]. *Physcomitrella* consisted mainly of dimer repeats [79]. In Brassicaceae, trinucleotide repeats were more common in the coding region, while other repeats were more common in the non-coding DNA [81]. Although the repetition types varied among different species, similarly, the number of SSRs in non-coding regions was usually greater than the number of coding regions.

### 4.4. Codon Usage

Most amino acids, except methionine and tryptophan, are encoded by 2–6 different codons [82]. Different species also use different synonymous codon. The main reason for this result is mutation and/or natural selection [83]. Although research on genetic mechanisms are limited, the selection of different codon can improve elongation and/or translation accuracy [84,85]. It has been found to affect protein expression, structure and function, as well as translation elongation [86].

The GC content of the third base of codon (GC_3_) is an important part of the GC content of genotype. GC content is an important indicator of the base composition of an organism’s genome. As the selection pressure on the third position of the codon is less than that on the first two, GC_3_ is often used as an indicator to measure the preference of codons. Dicotyledons tend to use A/U, while monocotyledons tend to use C/G [87,88]. The results showed that A/U was the main encoding with high frequency in *Trebouxiophyceae* sp. DW1.

### 4.5. Chloroplast Phylogenomics

Chloroplast phylogenomics has become an effective way to elucidate the mysterious evolutionary relationships at different taxonomic levels of plants [89]. The phylogenetic relationship of Chlorophyta, especially at higher taxonomic levels (order, class), has been a topic of debate. Therefore, we need to analyze more Trebouxiophyceae chloroplast genomes in order to reveal the deeper relationships of Trebouxiophyceae. There has been considerable debate regarding whether Trebouxiophyceae is a monophyly. Several studies showed that the Trebouxiophyceae is monophyletic [90,91], while others suggested that Trebouxiophyceae is paraphyletic [26,30,92]. Here, we used 36 protein-coding genes to reconstruct a more reliable phylogeny, which showed that Trebouxiophyceae is paraphyletic.

## Figures and Tables

**Figure 1 genes-13-01840-f001:**
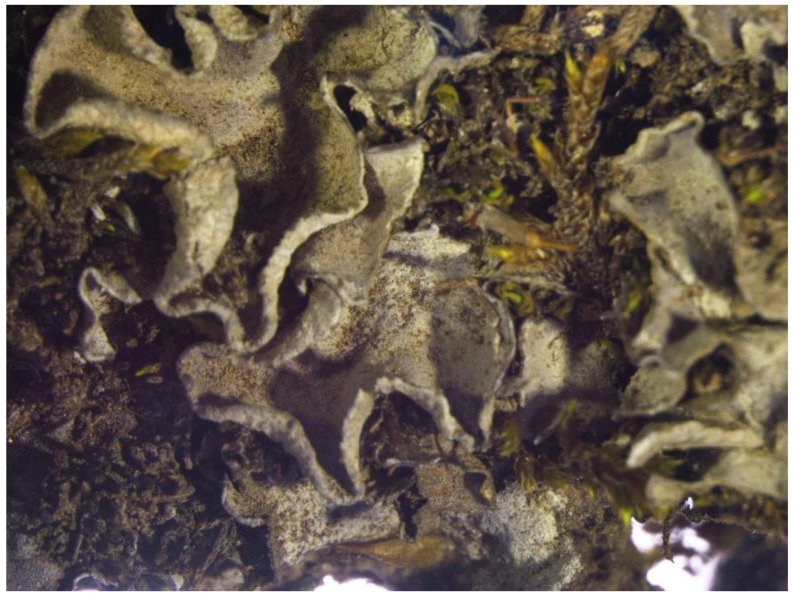
Morphological structure of *P. rufescens*.

**Figure 2 genes-13-01840-f002:**
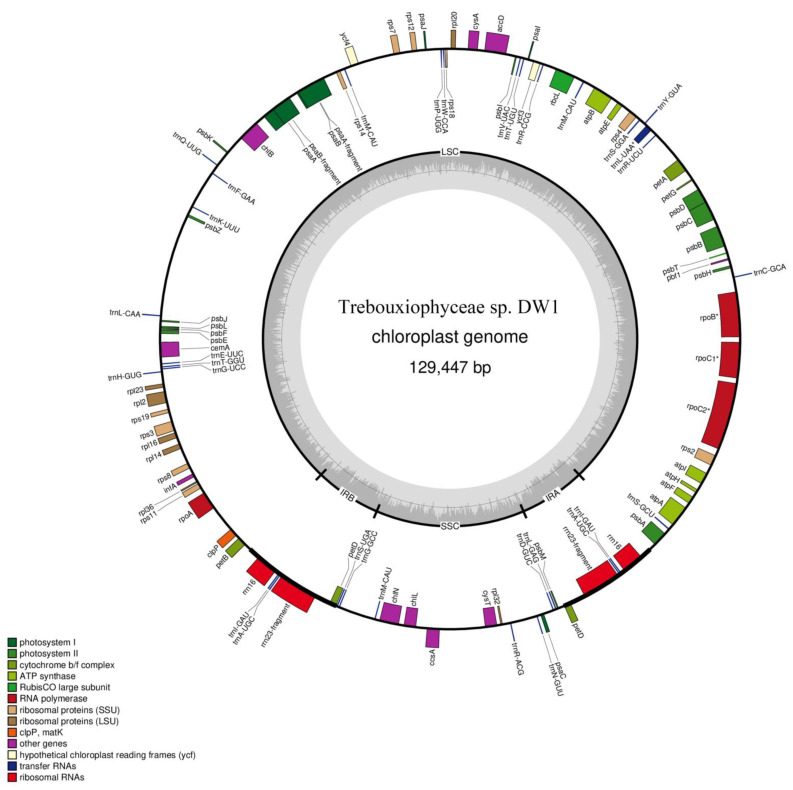
Gene map of the complete chloroplast genome of the green algae. *Trebouxiophyceae* sp. DW1. The genes inside the circle are transcribed clockwise, and the genes outside the circle are transcribed counterclockwise. Genes with introns are marked with an asterisk. The dark gray and light gray of the outer circle correspond to the LSC and SSC regions, respectively. The two regions are, respectively, represented as IRA and IRB, between LSC and SSC regions.

**Figure 3 genes-13-01840-f003:**
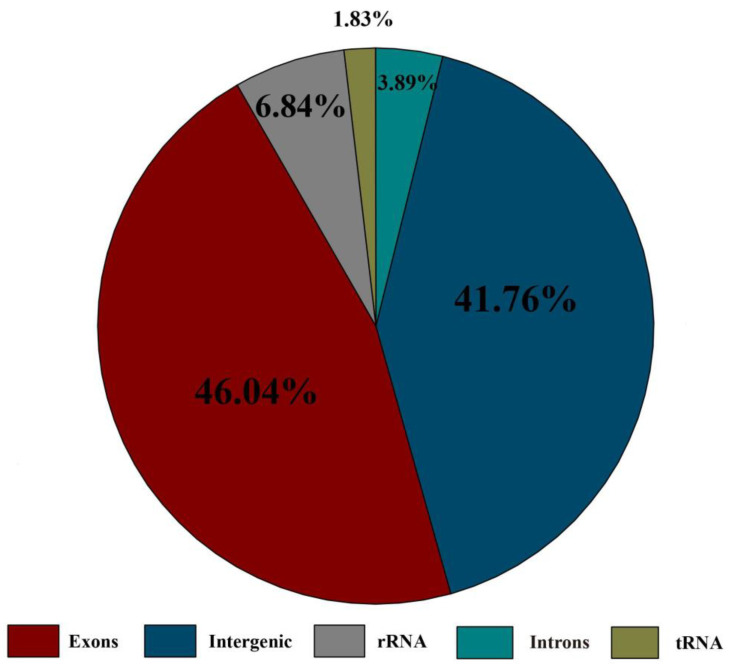
Proportions of the whole chloroplast genome of *Trebouxiophyceae* sp. DW1. The exons, introns, intergenic sequences, tRNA, and rRNA genes region.

**Figure 4 genes-13-01840-f004:**
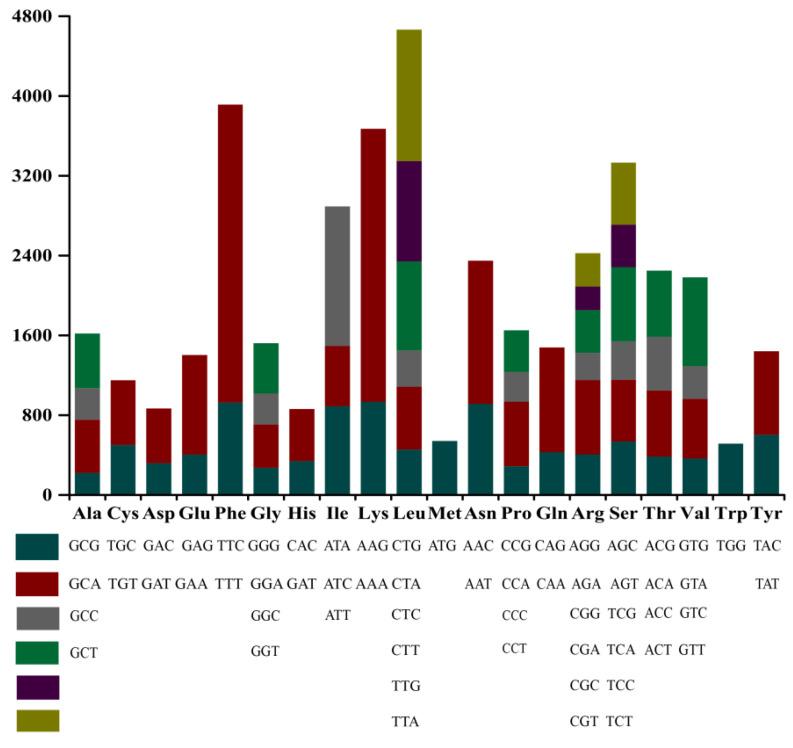
Codon usage in the whole chloroplast genome of *Trebouxiophyceae* sp. DW1.The *x*-axis represents amino acids, frequency of codon usage is plotted on the *y*-axis.

**Figure 5 genes-13-01840-f005:**
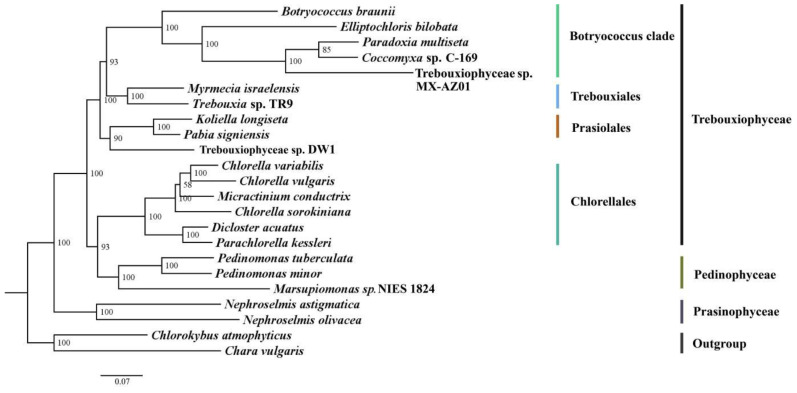
Maximum-likelihood (ML) phylogeny of 21 Trebouxiophyceae and Prasinophyceae species based on an alignment of 23,288 nt of 36 protein-coding genes. Bootstrap values are indicated in the nodes. The scale bar indicates substitutions/site.

**Figure 6 genes-13-01840-f006:**
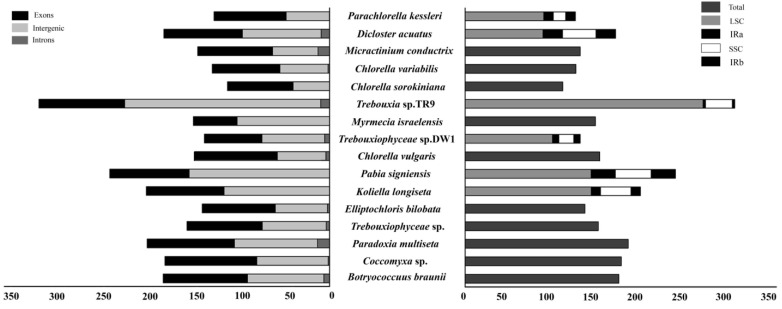
Total lengths of different regions of the cpDNAs in Trebouxiophyceae. The proportions of exons, introns and intergenic sequences in the chloroplast genome are shown on the left, and the proportions of IRS, LSC and SSC regions in the chloroplast genome are shown on the right.

**Table 1 genes-13-01840-t001:** Species and GenBank accession number used for phylogenetic analysis.

*Species*	*GenBank No.*
*Botryococcuus braunii*	*NC_025545*
*Coccomyxa sp.*	*NC_015084*
*Paradoxia multiseta*	*NC_025540*
*Trebouxiophyceae sp.*	*NC_018569*
*Elliptochloris bilobata*	*NC_025548*
*K. longiseta*	*NC_025531*
*P. signiensis*	*NC_025529*
*Chlorella vulgaris*	*NC_001565*
** *Trebouxiophyceae sp. DW1* **	** *MW_255987* **
*Myrmecia israelensis*	*NC_025525*
*Trebouxia sp. TR9*	*MK_643158*
*Chlorella sorokiniana*	*NC_023835*
*Chlorella variabilis*	*NC_015359*
*Micractinium conductrix*	*NC_036806*
*Dicloster acuatus*	*NC_025546*
*Parachlorella kessleri*	*NC_012978*
*Pedinomonas minor*	*NC_016733*
*Pedinomonas tuberculata*	*NC_025530*
*Marsupiomonas sp.*	*KM_462870*
*Nephroselmis astigmatica*	*NC_024829*
*Nephroselmis olivacea*	*NC_000927*
*Chara vulgaris*	*NC_008097*
*Chlorokybus atmophyticus*	*NC_008822*

Note: The new sequences generated in this study are indicated in boldface.

**Table 2 genes-13-01840-t002:** Repetitive sequences detected in the *Trebouxiophyceae* sp. DW1 chloroplast genome using the REPuter.

Number of Bases.	Type	Number
30–35	P	20
	F	14
36–40	P	7
	F	4
41–45	P	2
	F	1
46–50	P	1
	F	0

Note: forward repeats were denoted by F; palindromic repeats were denoted by P.

**Table 3 genes-13-01840-t003:** Tandem repeats detected in the *Trebouxiophyceae* sp. DW1 chloroplast genome using the Tandem Repeats Finder.

Indices	Period	Copy	Consensus	Percent	Percent	Score	A	C	G	T	Entropy
	Size	Number	Size	Matches	Indels						(0–2)
28,773–28,933	77	2.1	77	87	5	227	37	14	9	37	1.79
106,572–106,604	12	2.8	12	100	0	66	33	18	9	39	1.82

## Data Availability

The *Trebouxiophyceae* sp. DW1 isolated from the lichen *P. rufescens* are available from the corresponding author (reyim_mamut@xju.edu.cn) upon reasonable request. The lichen materials are available from the corresponding author upon request.

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
