# Peer review of "The Chloroplast Genome of the Lichen Photobiont Trebouxiophyceae sp. DW1 and Its Phylogenetic Implications"

_genes, 2022, doi:10.3390/genes13101840_

Round 1
Reviewer 1 Report
The manuscript: “The chloroplast genome of the lichen photobiont Trebouxiophyceae sp. DW1 and its phylogenetic implications” by Lidan Wang, Shenglu Zhang, Jinjin Fang, Xinjie Jin, Reyim Mamut and Pan Li, presents a genomic analysis for Trebouxiophyceae chloroplasts and comparisons with their recovered chloroplast genome for Trebouxiophyceae sp. DW1.
Main comments:
The introduction is complete and interesting.
The methods are, in general, well described and implemented.
Maybe you can include the k-mers used for the SPADES assembly.
Can you provide more information in section: 2.1. Culture conditions and isolation of phycobionts, to include the geographical site from which the sample comes. Also how the sample was collected and stored before the procedures?.
For the phylogenetic analysis, please specify the number of bootstrap repeats. It is not mention. For reference, the recommendation is to perform at least 1,000 generations, not 100.
In Table 1, I suggest including strains names for the species. In Supplementary Table 1, please correct typo “GrnBank”. Also please include a title or description for the supplementary files.
In Table 2, please correct “the.REPuter”. Please indicate the meaning of P and F in the table description. It would be more informative to include base coordinates for the repetitive sequences or the repeat sequences in the Supplementary Table 2.
In my opinion, Figure 3 is non-necessary as it provides the same information as Table 2.
For Figure 2, please add and space in “DW1.The”. Also I suggest: “The proportions of exons, introns, intergenic sequences, tRNA, and rRNA genes region are presented”. Also pie figures are not the best option to clearly present proportions to readers, and the figure is too big for the information it provides. Can you please change it to a bar plot.
In Figure 5, please add strain names for the species. You provide the complete name for your strain, so it would be advisable to include complete names for all strains. Also, please indicate that the bootstrap values are represented in percentage if you perform the recommended 1,000 bootstraps repeats. Finally please include a description for the scale bar (number of substitutions per length?)
Minor comments:
Why do you include the same information in tables and figures as supplementary material?
Would it be possible to include a short picture of Peltigera rufescens? For readers reference.
Line 216: please include strain name for Prasiolosis sp.
Line 254: please correct “is experiences”
Lines 262-263: can you further improve the phrase. “not a necessary part of the chloroplast energy of algae” is not clear enough.
Lines 265-267: can you further improve the writing of the sentence “SSRs are widespread (…) evolution and restructuring”.
Line 286: change “researches” to “research”
Line 291: there is a typo “conetent”, please correct
Lines 295-296: Lines 265-267: can you further improve the writing of the sentence (Our results show...)
Author Response
Response to Reviewer 1 Comment
Point 1: Maybe you can include the k-mers used for the SPADES assembly.
Response 1: k-mer=39
Point 2: Can you provide more information in section: 2.1. Culture conditions and isolation of phycobionts, to include the geographical site from which the sample comes. Also how the sample was collected and stored before the procedures?.
Response 2: The sample of P. rufescens was collected from Bayi Forest Farm, in Xinjiang Province. This voucher specimen was deposited in the Herbarium of College of Life Science and Technology at Xinjiang University in Urumchi, China, under the voucher number BY201830.
Point 3: For the phylogenetic analysis, please specify the number of bootstrap repeats. It is not mention. For reference, the recommendation is to perform at least 1,000 generations, not 100.
Response 3: bootstrap repeats were 1×106 generations
Point 4: In Table 1, I suggest including strains names for the species. In Supplementary Table 1, please correct typo “GrnBank”. Also please include a title or description for the supplementary files.
Response 4: This has been modified to GenBank
Point 5: In Table 2, please correct “the.REPuter”. Please indicate the meaning of P and F in the table description. It would be more informative to include base coordinates for the repetitive sequences or the repeat sequences in the Supplementary Table 2.
Response 5: P、F has been added, forward repeats was denoted by F; palindromic repeats was denoted by P.
Point 6: In my opinion, Figure 3 is non-necessary as it provides the same information as Table 2.
Response 6: Figure 3 has been deleted
Point 7: For Figure 2, please add and space in “DW1.The”. Also I suggest: “The proportions of exons, introns, intergenic sequences, tRNA, and rRNA genes region are presented”. Also pie figures are not the best option to clearly present proportions to readers, and the figure is too big for the information it provides. Can you please change it to a bar plot.
Response 7: Thank you very much for your suggestion. After our discussion, we still think that the pie chart is more intuitive.
Point 8: In Figure 5, please add strain names for the species. You provide the complete name for your strain, so it would be advisable to include complete names for all strains. Also, please indicate that the bootstrap values are represented in percentage if you perform the recommended 1,000 bootstraps repeats. Finally please include a description for the scale bar (number of substitutions per length?)
Response 8: Strain names of other species have been added; bootstrap values are indicated in the nodes. The scale bar indicates substitutions/site.
Point 9: Why do you include the same information in tables and figures as supplementary material?
Response 9: The supplementary file has been deleted with the same content as the article table
Point 10: Would it be possible to include a short picture of Peltigera rufescens? For readers reference.
Response 10: Peltigera rufescens picture has attached to the text
Point 11: Line 216: please include strain name for Prasiolosis sp.
Response 11: Modified to Prasiolosis sp.SAG 84.81
Point 12: Line 254: please correct “is experiences”
Response 12: Modified to is experienced
Point 13: Lines 262-263: can you further improve the phrase. “not a necessary part of the chloroplast energy of algae” is not clear enough.
Response 13: As mentioned in the previous sentence, the IRs of algae is smaller than that of higher plants and tends to degenerate gradually.
Point 14: Lines 265-267: can you further improve the writing of the sentence “SSRs are widespread (…) evolution and restructuring”.
Response 14: Modified to SSRs are widely found in prokaryotes and eukaryotes. They are inevitable and highly variable products of genome replication. They exist in coding and non-coding regions of the genome and play an important role in genome evolution and recombination.
Point 15: Line 286: change “researches” to “research”
Response 15: Modified to “research”
Point 16: Line 291: there is a typo “conetent”, please correct
Response 16: Modified to “content”
Point 17: Lines 295-296: Lines 265-267: can you further improve the writing of the sentence (Our results show...)
Response 17: Modified to the results showed that A/U was the main encoding with high frequency in Trebouxiophyceae sp. DW1.

Reviewer 2 Report
Comments on genes-1894609
Overall: This manuscript reports on a detailed analysis of the genome of the Trebouxiophyceae photobiont partner of the lichen forming fungus Peltigera rufescens.
The authors provide a complete gene map of the full genome and compliment this with an analysis of the exonic genes as well as the observance of repeat sequences.
The authors also describe the codon usage of this genome and also use the chloroplast genes to offer some insight to the evolutionary relationships with other related species.
Overall this work appears to be well executed and the results are consistent with other similar studies that exist in the literature.
This manuscript will be a most welcome addition to the literature as most of the work on lichen genomics has focussed on the fungal (mycobiont) partner. The addition of a genome analysis of the algal partner will be of broad interest to the lichen genomics community. The genomic information presented here will also help metagenomic sequencing efforts as it will help to serve as an additional reference genome for the algal partner.
This manuscript will be better once the minor comment below has been addressed.
Perhaps for a future publication (and not as additional work for this manuscript), the authors might consider scanning their genome for the presence of gene clusters associated with the biosynthesis of secondary metabolites (natural products). Some of the freely available annotation tools such as antiSMASH and MiBIG might reveal some illuminating information about the ability of the photobiont to produce secondary metabolites.
Specific Comments:
Section 2.4 Gene prediction: The Authors make the comment “and then the wrong genes were manually corrected by Sequin 101 software” however this analysis is completely unclear to a non-expert reader. Since this operation is central to the entire hypothesis of the paper it would be very helpful for the authors to include some more details here about what is meant by “wrong genes” and how they were “corrected”

Author Response
Response to Reviewer 2 Comment
Point 1: Section 2.4 Gene prediction: The Authors make the comment “and then the wrong genes were manually corrected by Sequin 101 software” however this analysis is completely unclear to a non-expert reader. Since this operation is central to the entire hypothesis of the paper it would be very helpful for the authors to include some more details here about what is meant by “wrong genes” and how they were “corrected”
Response 1: Modified to Sequin software manually adjust the position of start and stop codons.
